# Genetic Polymorphisms of MnSOD Modify the Impacts of Environmental Melamine on Oxidative Stress and Early Kidney Injury in Calcium Urolithiasis Patients

**DOI:** 10.3390/antiox11010152

**Published:** 2022-01-13

**Authors:** Chia-Chu Liu, Chia-Fang Wu, Yung-Chin Lee, Tsung-Yi Huang, Shih-Ting Huang, Hsun-Shuan Wang, Jhen-Hao Jhan, Shu-Pin Huang, Ching-Chia Li, Yung-Shun Juan, Tusty-Jiuan Hsieh, Yi-Chun Tsai, Chu-Chih Chen, Ming-Tsang Wu

**Affiliations:** 1Research Center for Environmental Medicine, Kaohsiung Medical University, Kaohsiung City 807, Taiwan; ccliu0204@gmail.com (C.-C.L.); cfwu27@nuu.edu.tw (C.-F.W.); u107800006@kmu.edu.tw (S.-T.H.); hsiehjun@kmu.edu.tw (T.-J.H.); 920254@kmuh.org.tw (Y.-C.T.); ccchen@nhri.edu.tw (C.-C.C.); 2Department of Urology, Kaohsiung Medical University Hospital, Kaohsiung Medical University, Kaohsiung City 807, Taiwan; 890197@kmuh.org.tw (Y.-C.L.); 970417@kmuh.org.tw (T.-Y.H.); shpihu@kmu.edu.tw (S.-P.H.); 850144@kmuh.org.tw (C.-C.L.); 840066@kmuh.org.tw (Y.-S.J.); 3Department of Urology, School of Medicine, College of Medicine, Kaohsiung Medical University, Kaohsiung City 807, Taiwan; 4Department of Urology, Pingtung Hospital, Ministry of Health and Welfare, Pingtung City 900, Taiwan; 5International Master Program of Translational Medicine, National United University, Miaoli 360, Taiwan; 6Department of Urology, Kaohsiung Municipal Siaogang Hospital, Kaohsiung City 812, Taiwan; 940199@kmuh.org.tw (H.-S.W.); 1030398@kmuh.org.tw (J.-H.J.); 7Graduate Institute of Medicine, College of Medicine, Kaohsiung Medical University, Kaohsiung City 807, Taiwan; 8Department of Internal Medicine, Divisions of Nephrology and General Medicine, Kaohsiung Medical University Hospital, Kaohsiung Medical University, Kaohsiung City 807, Taiwan; 9Department of Internal Medicine, School of Medicine, College of Medicine, Kaohsiung Medical University, Kaohsiung City 807, Taiwan; 10Division of Biostatistics and Bioinformatics, Institute of Population Health Sciences, National Health Research Institutes, Miaoli 350, Taiwan; 11Environmental and Occupational Medicine and Graduate Institute of Clinical Medicine, Kaohsiung Medical University, Kaohsiung City 807, Taiwan; 12Department of Family Medicine, Kaohsiung Medical University Hospital, Kaohsiung Medical University, Kaohsiung City 807, Taiwan; 13Department of Public Health, College of Health Sciences, Kaohsiung Medical University, Kaohsiung City 807, Taiwan

**Keywords:** manganese superoxide dismutase, genetic polymorphism, melamine, kidney injury, oxidative stress, calcium urolithiasis

## Abstract

Environmental melamine exposure increases the risks of oxidative stress and early kidney injury. Manganese superoxide dismutase (MnSOD), glutathione peroxidase, and catalase can protect the kidneys against oxidative stress and maintain normal function. We evaluated whether their single-nucleotide polymorphisms (SNPs) could modify melamine’s effects. A total of 302 patients diagnosed with calcium urolithiasis were enrolled. All patients provided one-spot overnight urine samples to measure their melamine levels, urinary biomarkers of oxidative stress and renal tubular injury. Median values were used to dichotomize levels into high and low. Subjects carrying the T allele of rs4880 and high melamine levels had 3.60 times greater risk of high malondialdehyde levels than those carrying the C allele of rs4880 and low melamine levels after adjustment. Subjects carrying the G allele of rs5746136 and high melamine levels had 1.73 times greater risk of high *N*-Acetyl-β-d-glucosaminidase levels than those carrying the A allele of rs5746136 and low melamine levels. In conclusion, the SNPs of MnSOD, rs4880 and rs5746136, influence the risk of oxidative stress and renal tubular injury, respectively, in calcium urolithiasis patients. In the context of high urinary melamine levels, their effects on oxidative stress and renal tubular injury were further increased.

## 1. Introduction

Melamine is a synthetic chemical used in manufacturing a variety of commercial daily life products including housewares, countertops, fabrics, glues, and flame retardants [1,2]. Because it has a high nitrogen content, it has been misused in animal feed and milk to deceptively elevate the protein content [1,2]. The adverse effects of melamine exposure drew worldwide attention after its scandalous addition to animal feeds, leading to the deaths of thousands of pet animals in the US in 2007, as well as to infant formula, resulting in urolithiasis in more than 50,000 children and six deaths in China in 2008 [1,2]. To date, melamine remains ubiquitously present in our environment. Several studies have detected melamine in water, soil, crops, daily food products, and animal tissues [3,4,5], and others detected it in most urine samples of several general populations from different countries [6,7,8,9]. 

After melamine intake, ninety percent of its original form will be excreted in urine within 24 h, so the kidneys may be more susceptible to melamine [1]. In addition to the effects of high doses of melamine on acute nephrotoxicity in children in 2008, long-term low-dose exposure to melamine has been linked to risk of kidney complications, including stone formation and deterioration of renal function in adults [10,11,12]. One probable mechanism that chronic low-dose melamine exposure could lead to early kidney injury and stone formation is its adverse effect on renal tubules, as was found by two of our human studies in melamine tableware workers [13] and adult patients with calcium urolithiasis [14]. We also conducted an in vitro study by using human renal proximal tubular HK-2 cells and found that melamine could induce renal tubular damage by increasing oxidative stress [15]. In two recent human studies, we also found that exposure to melamine increased the urinary biomarkers of oxidative stress, malondialdehyde (MDA), and 8-oxo-2′-deoxyguanosine (8-OHdG), with MDA mediating 36–53% of the total effect of melamine on a biomarker of renal tubular injury, *N*-Acetyl-β-d-glucosaminidase (NAG) [16]. Those findings suggest that exposure to low-dose environmental melamine might increase oxidative stress and further the risk of early kidney injury in humans.

Oxidative stress occurs when the generation of pro-oxidants or reactive oxygen species (ROS) exceeds the endogenous antioxidant capacity. Our kidneys are particularly sensitive to oxidative stress, which is thought to be an important factor in the initiation, development, and progression of most kidney diseases [17,18]. The cause of oxidative stress in kidneys may be associated with the interactions between medical diseases and environmental exposure to chemicals such as melamine [16]. Endogenous antioxidant systems including manganese superoxide dismutase (MnSOD), glutathione peroxidase (GPX1), and catalase (CAT) have been found to protect the kidneys against oxidative stress and subsequently help maintain normal function [18]. MnSOD can decompose toxic ROS, superoxide anions, to hydrogen peroxide, which is then converted to non-toxic water and oxygen by GPX1 and CAT in the mitochondria [18]. Several single-nucleotide polymorphisms (SNPs) of those antioxidant enzyme genes have been associated with various diseases [19], including kidney diseases [20,21,22]. 

Since environmental toxicant-associated kidney damage might be influenced by genetic factors [23], it is possible that the SNPs of antioxidant enzyme genes (e.g., *MnSOD*, *GPX1*, *CAT*) could modify the effects of environmental melamine exposure on the risk of oxidative stress and renal tubular injury in humans. To find out, we selected five candidate SNPs of antioxidant enzyme genes (*MnSOD:* rs4880 and rs5746136, *GPX1:* rs1800668, *CAT:* rs1001179 and rs769217), most of which have been linked to kidney diseases [20,21,22], to study each of their effects when combined with environmental melamine exposure on 8-OHdG and MDA, two biomarkers of oxidative stress, and NAG, a biomarker of renal tubular injury, in urine. 

## 2. Materials and Methods

### 2.1. Subjects

In total, 309 patients diagnosed with upper urinary tract calcium urolithiasis were enrolled between November 2010 and January 2015. The detailed study designs and protocols for their inclusion were described previously [14,16]. In brief, all patients were from Kaohsiung Medical University-affiliated hospitals in southwestern Taiwan. The eligible patients were individuals aged ≥ 20 years who had been diagnosed with urolithiasis in the upper urinary tract by radiography or ultrasonography, and who had provided stone specimens confirmed to have calcium components by infrared spectroscopy analysis (Spectrum RX I Fourier Transform-Infrared System, PerkinElmer, Waltham, MA, USA). None of the participants was found by X-ray to have radiolucent stones or by clinical evaluation to have uric acid or cystine stones. 

Patient subjects were excluded if they had a history of chronic urinary tract infection, chronic diarrhea, gout, hyperparathyroidism, renal tubular acidosis, renal failure, or cancer. Any subjects who had regularly taken vitamin D, calcium supplements, diuretics, or potassium citrate more than once per week within six months prior to the diagnosis of urolithiasis or interview were also excluded. The study protocol was approved by the Institutional Review Board of KMUH and all eligible patients provided signed informed written consent forms. This study followed the guidelines of STREGA [24] (see Appendix A).

### 2.2. Collection of Clinical Data and Biological Samples 

Upon admission, all participants provided blood and one-spot first-void morning urine samples after overnight fasting and before any treatment of urolithiasis for biochemical and genetic analyses. They also answered a structured questionnaire to collect their demographic data, medical history, and substance use (cigarette, alcohol, and betel quid) [14,16]. Participants were defined as cigarette smokers, alcohol drinkers, or betel quid chewers if they had regularly smoked ≥ 10 cigarettes per week, had consumed any alcoholic beverage ≥ 1 time per week, or had chewed ≥ 7 betel quids per week, respectively, for at least six months. Current users were those still practicing the above habits within one year before diagnosis of urolithiasis or the interview [11,25]. 

Clinical information, including stone location, stone number, stone diameter, and stone episodes, was also collected by questionnaire and further reviewed using the patients’ medical charts by one urologist (C.-C.L.) who was unaware of the exposure of interest, including melamine and biomarkers of oxidative stress (8-OHdG and MDA) and renal tubular injury (NAG) in urine and genotyping of antioxidant enzyme genes.

### 2.3. Analyses of Melamine, Biomarkers of Oxidative Stress (MDA and 8-OHdG), and Renal Tubular Injury (NAG) in Urine 

Urinary melamine was measured using an isotopic liquid chromatography–tandem mass spectrometry method (LC-MS/MS) (API4000Q, Applied Biosystems/MDS SCIEX, Concord, Vaughan, ON, Canada) [13]. Urinary MDA was measured using high-performance liquid chromatography with fluorescence (HPLC-FL) detection in a reversed-phase column (Luna C18, 250 × 4.6 nm) [26]. Urinary 8-OHdG was measured using a validated method for online solid-phase extraction (SPE) LC-MS/MS [27]. The detailed methods were described previously [14,16]. The limits of detection (LOD) for urinary melamine and biomarkers of oxidative stress were 0.4 ng/ml for melamine, 0.02 μmol/L for MDA, and 0.01 ng/mL for 8-OHdG. All urinary measurements of MDA and 8-OHdG were detectable. In contrast, 31 (10.0%) of the 309 urinary melamine measurements were below the LOD and were substituted as LOD/$\sqrt{2}$.

Urinary NAG was measured using an NAG assay kit (Diazyme Laboratory, Poway, CA, USA) [13,14]. All urinary NAG measurements were detectable. Urinary creatinine was analyzed by spectrophotometry (U-2000; Hitachi, Tokyo, Japan) set at a wavelength of 520 nm to measure the creatinine–picrate reaction [14]. The measurement of all biochemical parameters above were performed by two different laboratory technicians blinded to each other’s findings, study design, and participant information. 

### 2.4. Genotyping of Five SNPs

DNA was extracted from peripheral whole blood using a Puregene DNA Isolation Kit (Gentra Systems Inc., Minneapolis, MN, USA). The five SNPs (*MnSOD*-rs4880, *MnSOD*-rs5746136, *GPX1*-rs1800668, *CAT*-rs1001179, and *CAT*-rs769217) were analyzed using an assay-on-demand SNP genotyping kit for a TaqMan 5’ allelic discrimination assay (Applied Biosystems, Foster City, CA, USA). Briefly, SNP amplification assays including 10 ng of sample DNA in 25 μL of reaction solution contained 12.5 μL of the 2× TaqMan® Universal PCR Mix (Applied Biosystems), while 1.25 μL of pre-developed assay reagent from the SNP genotyping product (Applied Biosystems) contained two primers. Two MCB-Taqman probes were performed following the instructions of manufacturer. Polymerase chain reactions (PCRs) were performed using an ABI Prism 7500 Sequence Detection System (Applied Biosystems) [28,29]. These genotypes were confirmed by direct sequencing after full-scale genotyping. Randomly, ~10% of the study samples (30 cases) were repeated for quality control, with the results showing 100% accuracy.

### 2.5. Statistical Analyses

Quantitative data were expressed as means ± standard deviations (SD) or medians with interquartile ranges (IQR), and categorical data were presented as numbers (*n*) and percentages. The Hardy–Weinberg equilibrium for the distribution of genotypes was checked using the chi-square test. 

Urinary melamine levels and urinary biomarkers (MDA, 8-OHdG, NAG) were corrected by urinary creatinine values before further analyses. After correction, urinary biomarkers of oxidative stress (MDA and 8-OHdG) and renal tubular injury (NAG) were dichotomized into high or low using their median values [14,16]. Simple logistic regression models were first used to evaluate the associations between genotypes of antioxidant enzyme genes and urinary biomarkers of oxidative stress and renal tubular injury. If a significant relationship was noted in the initial analysis, the combined effects between genotypes of antioxidant enzyme genes and urinary melamine levels dichotomized by median values were further tested by multiple logistic regression analyses after adjusting for covariates, such as age, sex, BMI, educational level, personal habits, stone number, stone size, stone location, and comorbidities. All statistical analyses were performed using the SAS statistical package. All *p*-values were two-sided and considered significant if <0.05.

### 2.6. Sensitivity Analysis

To examine the robustness of our findings, we performed sensitivity analyses to compare the results when urinary melamine levels were divided into tertiles. Furthermore, we also compare the findings using a new method of covariate-adjusted standardization plus creatinine adjustment to correct for urine concentration and complicated confounding structures [14,30].

## 3. Results

Of the 309 study patients, 302 had data for genetic polymorphisms. There were 226 males and 76 females, with a mean age of 54.5 ± 12.9 years. In total, 156 participants (51.7%) presented first stone episodes and 194 (64.2%) presented single stones (Table 1). The median levels of the biomarkers in urine were 1.26 μg/mmol Cr for melamine, 0.24 μmol/mmol Cr for MDA, 5.78 mg/g Cr for 8-OHdG, and 0.86 IU/mmol Cr for NAG, which were the values used to dichotomize the levels into high and low for further analyses (Table 1). 

Five SNPs of antioxidant enzyme genes (*MnSOD*, *GPX1*, and *CAT*) are shown in Table 2, all of which conformed to the Hardy–Weinberg equilibrium. Because the rare homozygotes of both GPX1-rs1800668 and CAT-rs1001179 had only one case, they were combined with heterozygotes for the subsequent analyses (Table 3 and Table 4 and Appendix A). 

In *MnSOD*-rs4880, subjects who carried the *T* allele had a significantly higher risk of having high MDA levels than those carrying the *C* allele (odds ratio (OR) = 1.80, 95% CI = 1.11–2.92) (Table 3). In *MnSOD*-rs5746136, subjects who carried *GG* genotype had 1.99-fold higher risk of having high NAG levels (95% CI = 1.00–3.99) compared to those with *AA* genotype (Table 4). There were no significant relationships between five SNPs of antioxidant enzyme genes and 8-OHdG (Appendix A). Sensitivity analyses using a new method of covariate-adjusted standardization plus creatinine adjustment found similar results (Appendix A).

Table 5 shows the effects of *MnSOD*-rs4880 combined with high and low urine melamine levels on the risk of high MDA levels (≥50% vs. <50%). Subjects who carried the *T* allele and had high melamine levels had a significantly higher risk of high MDA levels than those who carried the *C* allele and had low melamine levels (adjusted OR = 3.60, 95% CI = 1.79–7.22), after adjusting for age, sex, BMI, educational level, personal habits, clinical stone characteristics, and comorbidities. Dichotomized urinary MDA levels to <66.6% and ≥66.6%, the results remained the same (Table 5). The results also remained similar when we used a new method of covariate-adjusted standardization plus creatinine adjustment (Appendix A).

Table 6 shows the effects of *MnSOD*-rs5746136 combined with urine melamine levels on the risk of high NAG levels (≥50% vs. <50%). Subjects who carried the *G* allele and had high melamine levels had a significantly higher risk of high NAG levels than those who carried the *A* allele and had low melamine levels (adjusted OR = 1.73, 95% CI = 1.04–2.89) in the fully adjusted model. Dichotomized urinary NAG levels to <66.6% and ≥66.6%, the results remained consistent (Table 6). The results also remained similar when we used a new method of covariate-adjusted standardization plus creatinine adjustment (Appendix A).

## 4. Discussion

Increased risk of oxidative stress and renal tubular injury was previously found in adult patients with calcium urolithiasis exposed to environmental melamine [14,16]. In that same population, the current study found that two SNPs of the antioxidant enzyme gene, *MnSOD*, modified the effect of melamine exposure on those risks. In conjunction with high urinary melamine levels (≥50%), subjects carrying *T* allele of *MnSOD*-rs4880 were at even greater risk of high urinary MDA levels, a biomarker of oxidative stress, compared to its *C* allele carriers (adjusted OR = 3.60, 95% CI = 1.79–7.22 vs. 1.65, 95% CI = 0.62–4.37) (Table 5). Furthermore, subjects carrying *G* allele of *MnSOD*-rs5746136 were also at even greater risk of high urinary NAG levels, a biomarker of renal tubular injury, than its *A* allele carriers (adjusted OR = 1.73, 95% CI = 1.04–2.89 vs. 1.02, 95% CI = 0.59–1.77) (Table 6).

Melamine is widely used in the production of laminates, plastics, glues, adhesives, and coatings, and its derivatives are also used in flame retardants and in insulation [31]. Because of its resilience, light weight, and low cost, melamine is also widely used as a substitute for porcelain in the production of tableware and food utensils [31]. Our early studies found that melamine-made tableware could leach substantial amounts of melamine, especially in highly acidic soups or soups served at high temperatures, making them a major source of environmental melamine exposure [32,33]. In addition to our own in vitro study using human renal proximal tubular HK-2 cells, several studies have found that melamine could increase ROS and induce injury in other kidney cell lines, including a rat kidney epithelial cell line (NRK-52e cells) [34] and a human embryonic kidney cell line [35]. Furthermore, several extrinsic antioxidants, such as catechin [36] and bee honey [37], have been found to reverse melamine-induced ROS and prevent further nephrotoxicity in animal models. 

The current study found that two SNPs of *MnSOD*, rs4880 and rs5746136, modified the effects of melamine exposure on the risk of oxidative stress and renal tubular injury, respectively. MnSOD is a key enzyme found in our antioxidant defense systems and the most important member of the superoxide dismutase family known to play a crucial role in controlling mitochondrial superoxide radicals [18]. Oxidative stress in the mitochondria has been found to play a vital role in the pathogenesis of kidney disease and injury [18]. *MnSOD*-rs 4880 is present in exon 2 and substitutes a *C > T* at position 2734, changing the amino acid from alanine (Ala) to valine (Val) at position 16 [19,20]. The presence of the *T* (Val) allele leads to the production of instable mRNA and reduces the transport of the enzyme into the mitochondrial matrix, reducing its antioxidant function [19]. This study found that subjects who carried the *T* allele had a significantly higher risk of high urinary MDA, a biomarker of oxidative stress, than those who carried the *C* allele. In a recent case–control study of 256 patients with end-stage renal disease (ESRD) underlying hemodialysis and 374 controls, subjects with the *TT* genotype were found to have significantly higher serum MDA levels than those with the *CC* genotype (2.57 ± 0.79 vs 2.17 ± 0.78 mmol/L, *p* < 0.05) [38]. That study also found the *TT* genotype to be an independent risk factor for the development of ESRD [38]. In a prospective cohort study of 185 patients with chronic kidney disease followed for up to 12 months, subjects with *TT* and *CT* genotypes were found to have a greater decline in kidney function than those with the *CC* genotype [20]. In another prospective cohort study (SURGENE study) of 340 patients with type 1 diabetes followed for up to ten years, the *T* allele was associated with a higher incidence of renal events (new cases of incipient nephropathy or the progression to a more severe stage of nephropathy) and with a decline in estimated glomerular filtration rate during follow-up [39]. 

*MnSOD*-rs5746136 is located at 65 bp downstream from the poly(A) site and near the SP1 and the NF-kB transcription element sequences, which may play a role in the regulation of *MnSOD* gene expression [40]. The current study found that subjects who carried the *G* allele of *MnSOD*-rs5746136 had a significantly higher risk of high urinary NAG, a biomarker of renal tubular injury, than those carrying the *A* allele. This SNP has also been linked to several diseases associated with oxidative stress. In a recent case–control study of 100 Chinese children with Kawasaki disease and 102 healthy controls, subjects with the *A* allele had a 0.558-times lower risk (95% CI = 0.371–0.838) of Kawasaki disease than those with the *G* allele. In another case–control study of 164 Turkish patients with polycystic ovary syndrome and 148 healthy controls, subjects with *GG* and *AG* genotypes had a 2.95-fold increase in risk (95% Cl: 1.2–3.1) of having polycystic ovary syndrome compared those with the AA genotype [41]. However, in the SURGENE study, which followed 340 Caucasian patients with type 1 diabetes for up to ten years, subjects with the *G* allele were reported to have a significantly lower risk of having established or advanced nephropathy compared to those with the *A* allele (OR:0.30, 95% GI = 0.11–0.72) at the end of follow-up [39]. Further studies are needed to elucidate the role of *MnSOD*-rs5746136 on the risk of kidney diseases, including urolithiasis formation and the deterioration of renal function in populations of different clinical and ethnic backgrounds. 

Traditionally, melamine was considered a relatively safe chemical. After the toxic milk and food scandals, the WHO and US FDA lowered their recommended tolerable daily intake (TDI) of melamine to 200 and 63 μg per kg body weight per day (μg/kg_bw/day), respectively [31]. Recently, we evaluated the one-sided 95% lower bound of benchmark dose (BMDL) of melamine exposure, taking into consideration a benchmark response of 0.10 in two vulnerable human populations, calcium urolithiasis patients and early chronic kidney disease patients [31,42]. It was concluded that both BMDLs should be lowered to 4.89 μg/kg_bw/day and 0.74 to 2.03 μg/kg_bw/day, respectively, to protect calcium urolithiasis patients and early chronic kidney disease patients from further deterioration of renal function [31,42]. Both simulated BMDLs for melamine exposure threshold in those susceptible populations were much lower than the current WHO and the US FDA recommended TDIs. These findings were supported by a toxicological analysis reporting that a lower TDI of melamine to 8.1 μg/kg_bw/day should be considered for general population [43]. This study found that two SNPs of *MnSOD*, rs4880 and rs5746136, modify the risks of oxidative stress and renal tubular injury, respectively. Furthermore, they strengthen the risks of melamine on high urinary biomarkers of oxidative stress (MDA) and renal tubular injury (NAG), respectively. Our findings support the hypothesis that genetic factors can influence melamine-induced oxidative stress and kidney damage. Therefore, in the future, genetic factors may need to be taken into consideration when assessing the human TDI, especially in vulnerable populations. In addition, it is important to prevent environmental melamine exposure, for example by avoiding the use of melamine-made tableware, especially when containing highly acidic or high-temperature soups [32,33]. A behavior intervention to decrease the use of melamine-made tableware was found to effectively prevent melamine exposure from the environment [44]. Other studies have also reported that natural products such as catechin [36] and bee honey [37] could reverse melamine-induced ROS and prevent further nephrotoxicity. The effects of other natural product or plant extract supplementation are warranted for further investigation [45,46]. 

This study has some limitations. First, we only measured melamine levels based on one-spot first-void morning urine samples. Although we previously found good correlations between melamine levels in one-spot first-void morning urine and total melamine excretion in previous 24 h urine [7,11], these measurements might not completely represent the long-term melamine exposure of participants. Second, we did not evaluate melamine analogues, especially cyanuric acid, or other environmental nephrotoxins, such as lead and cadmium. Further studies might need to elucidate their possible synergistic effects on the adverse outcomes of kidney. Third, we did not evaluate SNPs of other *SOD* genes such as Cu/ZnSOD and their impacts on mitochondrial function. Further studies might need to elucidate their possible relationships. 

## 5. Conclusions

Two SNPs, rs4880 and rs5746135, of *MnSOD* were found to modify the risks of oxidative stress and renal tubular injury, respectively, in adult patients with calcium urolithiasis. Their presence increased the risk already posed by high urinary melamine levels in the same population, supporting the hypothesis that genetic factors can influence melamine-induced oxidative stress and kidney damage. Future assessments may want to factor in these SNPs by setting a TDI for melamine, especially in vulnerable populations. Further studies of different ethnic populations are needed to confirm our preliminary results.

## Figures and Tables

**Table 1 antioxidants-11-00152-t001:** Characteristics and laboratory data in 302 calcium urolithiasis patients.

Variables	*N* (%)
Gender Male Female	226 (74.8) 76 (25.2)
Years of education ≤9 >9	143 (47.4) 159 (52.6)
Personal habits	
Current smokers	104 (34.4)
Current betel quid chewers	21 (7.0)
Current drinkers	41(13.6)
Diabetes mellitus	52 (17.2)
Hypertension	105 (34.8)
Dyslipidemia	21 (7.0)
Stone episodes	
1 ≥2	156 (51.7) 146 (48.3)
Stone location	
Kidney Ureter Kidney and Ureter	69 (22.8) 181 (59.9) 52 (17.2)
Stone number Single Multiple	194 (64.2) 108(35.8)
Maximal stone diameter <1 cm	184 (60.9)
1–2 cm >2 cm	55 (18.2) 63 (20.9)
	**Mean ± SD (median, IQR)**
Age (years)	54.5 ± 12.9 (55.0, 46.8–63.0)
BMI (kg/m^2^)	26.3 ± 4.0 (25.9, 23.8–28.4)
Urinary biomarkers	
Melamine (ng/mL)	12.47 ± 29.51 (4.86, 1.94–12.67)
MDA (μmol/L)	1.16 ± 0.86 (0.93, 0.52–1.60)
8-OHdG (ng/mL)	3.30 ± 2.46 (2.59, 1.55–4.46)
NAG (IU/L)	5.23 ± 5.62 (3.77, 1.90–6.33)
Urine biomarkers corrected by urine creatinine	
Melamine (μg/mmol Cr)	3.27 ± 6.68 (1.26, 0.48–3.34)
MDA (μmol/mmol Cr)	0.31 ± 0.28 (0.24, 0.15–0.37)
8-OHdG (mg/g Cr)	7.82 ± 7.41 (5.78, 4.08–9.11)
NAG (IU/mmol Cr)	1.36 ± 1.52 (0.86, 0.55–1.60)
Urinary biomarkers corrected by method of covariate-adjusted standardization plus covariate adjustment	
Melamine (ng/mL)	12.59 ± 26.10 (4.89, 1.94–12.76)
MDA (μmol/L)	1.18 ± 1.05 (0.91, 0.58–1.37)
8-OHdG (ng/mL)	3.37 ± 3.18 (2.57, 1.70–3.81)
NAG (IU/L)	5.12 ± 5.80 (3.14, 2.09–6.06)

Abbreviations: *N* = number; SD = standard deviation; IQR = interquartile range; BMI = Body mass index; MDA = malondialdehyde; 8-OHdG = 8-oxo-2’-deoxyguanosine; NAG = *N*-acetyl b-d-glucosaminidase; Cr = creatinine.

**Table 2 antioxidants-11-00152-t002:** Distribution of SNPs of antioxidant enzyme genes (*MnSOD, GPX1,* and *CAT*) in 302 calcium urolithiasis patients.

	Alleles	*N* (%)	Genotypes	*N* (%)
*MnSOD*-rs4880	*C*	80 (13.2)	*CC*	8 (2.6)
	*T*	524 (86.8)	*TC*	64 (21.2)
			*TT*	230 (76.2)
*MnSOD*-rs5746136	*A*	238 (39.4)	*AA*	48 (15.9)
	*G*	366 (60.6)	*AG*	142 (47.0)
			*GG*	112 (37.1)
*GPX1*-rs1800668	*T*	21 (3.5)	*TT*	1 (0.3)
	*C*	583 (96.5)	*TC*	19 (6.3)
			*CC*	282 (93.4)
*CAT*-rs1001179	*T*	17 (2.8)	*TT*	1 (0.3)
	*C*	587 (97.2)	*TC*	15 (5.0)
			*CC*	286 (94.7)
*CAT*-rs769217	*T*	289 (47.8)	*TT*	65 (21.5)
	*C*	315 (52.2)	*TC*	159 (52.6)
			*CC*	78 (25.9)

Abbreviations: SNP = single nucleotide polymorphism; *MnSOD* = manganese superoxide dismutase; *GPX1* = glutathione peroxidase; *CAT* = catalase; *N* = number.

**Table 3 antioxidants-11-00152-t003:** Relationships of SNPs of antioxidant enzyme genes with a urinary biomarker of oxidative stress, MDA, in 302 calcium urolithiasis patients.

		MDA, *N* (%)			
*MnSOD*-rs4880		Low < 50%	High ≥ 50%	Crude OR (95% CI)	*p* Value
Alleles	*C*	50 (16.6)	30 (9.9)	Ref	
	*T*	252 (83.4)	272 (90.1)	1.80 (1.11–2.92)	0.017
Genotypes	*CC*	5 (3.3)	3 (2.0)	Ref	
	*CT*	40(26.5)	24 (15.9)	1.00 (0.22-4.56)	1
	*TT*	106 (70.2)	124 (82.1)	1.95 (0.46-8.35)	0.368
	*p* for trend				0.023
***MnSOD*-rs5746136**					
Alleles	*A*	118 (39.1)	120 (39.7)	Ref	
	*G*	184 (60.9)	182 (60.3)	0.97 (0.73–1.35)	0.868
Genotypes	*AA*	27 (17.9)	21 (13.9)	1	
	*AG*	64 (42.4)	78 (51.7)	1.57 (0.81–3.03)	0.182
	*GG*	60 (39.7)	52 (34.4)	1.11 (0.56–2.20)	0.755
	*p* for trend				0.869
***GPX1*-rs1800668**					
Alleles	*T*	7 (2.3)	14 (4.6)	Ref	
	*C*	295 (97.7)	288 (95.4)	0.49 (0.19–1/23)	0.127
Genotypes	*TT + TC*	7 (4.6)	13 (8.6)	Ref	
	*CC*	144 (95.4)	138 (91.4)	0.52 (0.20–1.33)	0.171
***CAT*-rs1001179**					
Alleles	*T*	7 (2.3)	10 (3.3)	Ref	
	*C*	295 (97.7)	292 (96.7)	0.69 (0.26–1.85)	0.463
Genotypes	*TT + TC*	6 (4.0)	10 (6.6)	Ref	
	*CC*	145 (96.0)	141 (93.4)	0.58 (0.21–1.65)	0.309
***CAT*-rs769217**					
Alleles	*T*	146 (48.3)	143 (47.4)	Ref	
	*C*	156 (51.7)	159 (52.6)	1.04 (0.76–1.63)	0.807
Genotypes	*TT*	35 (23.2)	30 (19.9)	Ref	
	*TC*	76 (50.3)	83 (55.0)	1.27 (0.71–2.27)	0.412
	*CC*	40 (26.5)	38 (25.1)	1.11 (0.57–2.14)	0.76
	*p* for trend				0.802

Abbreviations: SNP = single nucleotide polymorphism; MDA = malondialdehyde; *N* = number; *MnSOD* = manganese superoxide dismutase; *GPX1* = glutathione peroxidase; *CAT* = catalase.

**Table 4 antioxidants-11-00152-t004:** Relationships of SNPs of antioxidant enzyme genes with a urinary biomarker of renal tubular injury, NAG, in 302 calcium urolithiasis patients.

		NAG, *N* (%)			
*MnSOD*-rs4880		Low < 50%	High ≥ 50%	Crude OR (95% CI)	*p* Value
Alleles	*C*	42 (13.9)	38 (12.6)	Ref	
	*T*	260 (86.1)	264 (87.4)	1.12 (0.70–1.80)	0.631
Genotypes	*CC*	4 (2.6)	4 (2.6)	Ref	
	*CT*	34 (22.5)	30 (19.9)	0.88 (0.20–3.84)	0.87
	*TT*	113 (74.9)	117 (77.5)	1.04 (0.25–4.24)	0.96
	*p* for trend				0.898
***MnSOD*-rs5746136**					
Alleles	*A*	130 (43.0)	108 (35.8)	Ref	
	*G*	172 (57.0)	194 (64.2)	1.36 (0.98–1.88)	0.067
Genotypes	*AA*	30(19.8)	18 (11.9)	Ref	
	*AG*	70 (46.4)	72 (47.7)	1.71 (0.88–3.35)	0.115
	*GG*	51 (33.8)	61 (40.4)	1.99 (1.00–3.99)	0.051
	*p* for trend				0.07
***GPX1*-rs1800668**					
Alleles	*T*	11 (3.6)	10 (3.3)	Ref	
	*C*	291 (96.4)	292 (96.7)	1.10 (0.46–2.64)	0.824
Genotypes	*TT + TC*	11 (7.3)	9 (6.0)	Ref	
	*CC*	140 (92.7)	142 (94.0)	1.24 (0.50–3.08)	0.64
***CAT*-rs1001179**					
Alleles	*T*	9 (3.0)	8 (2.6)	Ref	
	*C*	293 (97.0)	294 (97.4)	1.13 (0.43–2.97)	0.806
Genotypes	*TT + TC*	8 (5.3)	8 (5.3)	Ref	
	*CC*	143 (94.7)	143 (94.7)	1 (0.37–2.74)	1
***CAT*-rs769217**					
Alleles	*T*	137 (45.4)	152 (50.3)	Ref	
	*C*	165 (54.6)	150 (49.7)	0.82(0.60–1.13)	0.222
Genotypes	*TT*	34 (22.5)	31 (20.5)	Ref	
	*TC*	69 (45.7)	90 (59.6)	1.43(0.80–2.55)	0.225
	*CC*	48 (31.8)	30 (19.9)	0.69(0.35–1.34)	0.267
	*p* for trend				0.21

Abbreviations: SNP = single-nucleotide polymorphism; NAG = *N*-acetyl b-d-glucosaminidase; *N* = number; *MnSOD* = manganese superoxide dismutase; *GPX1* = glutathione peroxidase; *CAT* = catalase.

**Table 5 antioxidants-11-00152-t005:** Combined effects of *MnSOD* genetic polymorphism (rs4880) and urine melamine levels on the risk of a high urinary marker of oxidative stress, MDA, in 302 calcium urolithiasis patients.

		**MDA, *N* (%)**				**Model 1**		**Model 2**	
**rs4880 and Melamine**	** *N* **	**Low** **<50%**	**High** **≥50%**	**Crude OR** **(95% CI)**	***p* Value**	**Adjusted OR (95% CI)**	***p* Value**	**Adjusted OR (95% CI)**	***p* Value**
*C* allele + Melamine < 50%	48	32 (10.6)	16 (5.3)	1		1		1	
*C* allele + Melamine > 50%	32	18 (6.0)	14 (4.6)	1.56 (0.62–3.91)	0.347	1.70 (0.64–4.49)	0.286	1.65 (0.62–4.38)	0.317
*T* allele + Melamine < 50%	254	156 (51.6)	98 (32.5)	1.26 (0.66–2.41)	0.492	1.28 (0.64–2.56)	0.484	1.28 (0.64–2.56)	0.491
*T* allele + Melamine > 50%	270	96 (31.8)	174 (57.6)	3.63 (1.89–6.94)	<0.001	3.64 (1.82–7.27)	<0.001	3.60 (1.79–7.22)	<0.001
*p* for trend					<0.001		<0.001		<0.001
		**MDA, *N* (%)**				**Model 1**		**Model 2**	
**rs4880 and Melamine**	** *N* **	**Low** **<66.6%**	**High** **≥66.6%**	**Crude OR** **(95% CI)**	***p* Value**	**Adjusted OR (95% CI)**	***p* Value**	**Adjusted OR (95% CI)**	***p* Value**
*T* allele + Melamine < 50%	48	37 (9.2)	11 (5.5)	1		1		1	
*T* allele + Melamine > 50%	32	22 (5.4)	10 (5.0)	1.53 (0.56–4.18)	0.408	1.64 (0.57–4.73)	0.365	1.61 (0.55–4.68)	0.384
*C* allele + Melamine < 50%	254	185 (45.8)	69 (34.5)	1.26 (0.61–2.60)	0.541	1.28 (0.59–2.74)	0.534	1.28 (0.59–2.76)	0.528
*C* allele + Melamine > 50%	270	160 (39.6)	110 (55.0)	2.31 (1.13–4.73)	0.022	2.19 (1.03–4.65)	0.042	2.20 (1.03–4.70)	0.042
*p* for trend					0.002		0.009		0.01

Abbreviations: MnSOD = manganese superoxide dismutase; MDA = malondialdehyde; *N* = number. Model 1: Adjusting for age, sex, BMI, educational level, alcohol drinking, cigarette smoking, betel quid chewing, stone number, stone location, and stone size. Model 2: Adjusting for age, sex, BMI, educational level, alcohol drinking, cigarette smoking, betel quid chewing, stone number, stone location, stone size, hypertension, diabetes mellitus, and dyslipidemia.

**Table 6 antioxidants-11-00152-t006:** Combined effects of *MnSOD* genetic polymorphism (rs5746136) and urine melamine levels on the risk of a high urinary marker of renal tubular injury, NAG, in 302 calcium urolithiasis patients.

		**NAG, *N* (%)**				**Model 1**		**Model 2**	
**rs5746136 and Melamine**	** *N* **	**Low < 50%**	**High** **≥50%**	**Crude OR** **(95% CI)**	***p* Value**	**Adjusted OR (95% CI)**	***p* Value**	**Adjusted OR (95% CI)**	***p* Value**
*A* allele + Melamine < 50%	110	62 (20.5)	48 (15.9)	1		1		1	
*A* allele + Melamine > 50%	126	66 (21.9)	60 (19.9)	1.14 (0.68–1.90)	0.617	1.07 (0.62–1.83)	0.808	1.02 (0.59–1.77)	0.931
*G* allele + Melamine < 50%	192	100 (33.1)	92 (30.4)	1.21 (0.76–1.94)	0.424	1.11 (0.68–1.82)	0.68	1.11 (0.67–1.82)	0.689
*G* allele + Melamine > 50%	176	74 (24.5)	102 (33.8)	1.78 (1.10–2.88)	0.019	1.78 (1.08–2.95)	0.025	1.73 (1.04–2.89)	0.036
*p* for trend					0.019		0.024		0.029
		**NAG, *N* (%)**				**Model 1**		**Model 2**	
**rs5746136 and Melamine**	** *N* **	**Low < 66.6%**	**High** **≥66.6%**	**Crude OR** **(95% CI)**	***p* Value**	**Adjusted OR (95% CI)**	***p* Value**	**Adjusted OR (95% CI)**	***p* Value**
*A* allele + Melamine < 50%	110	80 (19.9)	30 (14.9)	1		1		1	
*A* allele + Melamine > 50%	126	88 (21.9)	38 (18.8)	1.13 (0.64–1.98)	0.681	1.05 (0.58–1.89)	0.877	1.01 (0.55–1.82)	0.991
*G* allele + Melamine < 50%	192	132 (32.8)	60 (29.7)	1.23 (0.73–2.07)	0.433	1.12 (0.66–1.92)	0.67	1.12 (0.65–1.93)	0.678
*G* allele + Melamine > 50%	176	102 (25.4)	74 (36.6)	1.94 (1.16–3.24)	0.012	1.86 (1.09–3.17)	0.024	1.78 (1.03–3.07)	0.038
*p* for trend					0.008		0.016		0.023

Abbreviations: *MnSOD* = manganese superoxide dismutase; NAG = *N*-acetyl b-d-glucosaminidase; *N* = number. Model 1: Adjusting for age, sex, BMI, educational level, alcohol drinking, cigarette smoking, betel quid chewing, stone number, stone location, and stone size. Model 2: Adjusting for age, sex, BMI, educational level, alcohol drinking, cigarette smoking, betel quid chewing, stone number, stone location, stone size, hypertension, diabetes mellitus, and dyslipidemia.

## Data Availability

Data are contained in the article and Appendix A.

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
