# Peer review of "Genetic Polymorphisms of MnSOD Modify the Impacts of Environmental Melamine on Oxidative Stress and Early Kidney Injury in Calcium Urolithiasis Patients"

_antioxidants, 2022, doi:10.3390/antiox11010152_

Round 1
Reviewer 1 Report
Comments to Authors
In this MS entitled ‘Genetic polymorphisms of MnSOD modify the impact of envi-2 ronmental melamine on oxidative stress and early kidney injury in calcium urolithiasis patients’ Chia-Chu Liu et al., reported that genetic polymorphism in MnSOD influence the risk of oxidative stress and renal tubular injury, respectively, in calcium urolithiasis patients. This is well thought, organized and well-presented manuscript with some clinically significant information. Still there are some minor concern and hopefully addressing those point will strengthen this manuscript unfoundedly.
Introduction section need some information
First sentence is not clear reader would prefer to know what these daily life products are.
Rationale of the study is not clear why Melanine target kidney (nephrotoxicity) and what cells in kidney are the primary target and why.
Much interest is focused on MnSOD where as Cu/Zn SOD is completely ignored in this study but explanation is not given. If MnSOD is critical then some mitochondrial marker should also be studied for perturbed functionality.
Is one-time urine sampling collection appropriate to make the final decision? I am aware that authors mentioned that it is the limitation of the study which dilute the significance of all these observations and need a clear explanation.
Subject age on page 3 is > 20 whereas in result section second line is 54.5+12.9 this is little confusing.
Are there any patients who represent SNP for all three antioxidant enzymes? As stated in conclusion that Two SNPs, rs4880 and rs5746135, of MnSOD were found to modify the risk of oxidative stress and renal tubular injury, respectively. These results suggest that Glutathione peroxidase and Catalase are not prominent antioxidant enzymes here.
Statistical analysis is difficult to understand, is there any significance of difference.
Check authors list, last word and should be deleted.
Reviewer 2 Report
The Research work entitled, 'Genetic polymorphisms of MnSOD modify the impact of environmental melamine on oxidative stress and early kidney injury in calcium uro-lithiasis patients' is a good work done by C C Liu and colleagues. The research parameters evaluated are excellent and methodology followed is also good.
The abstract written fairly well, Introduction and methodologies are well written too.
In the results section there are some suggestions to be incorporated
Table legends are not clear, the author did not describe anywhere whether the values are in Mean+/- SEM or not.
The authors could have presented results in graphs why the raw data provided.
The discussion is not satisfactory authors must elaborate and compare the recent research outcomes with their results.
Conclusion is enough good.
The manuscript can be considered for publication after the corrections suggested
Reviewer 3 Report
The authors deal with Genetic polymorphisms of MnSOD modify the impact of environmental melamine on oxidative stress and early kidney injury in calcium urolithiasis patients. The article is well constructed and gives important information regarding the interaction between genotype and environmental exposure to melamine related to the level of oxidative stress and kidney injury. The novelty of the article is important and also its interdisciplinary nature. The methodology is complex and ample, presenting the numerous investigation modalities such as liquid chromatography, tandem mass spectrometry method, genotyping.
The Discussion chapter can be improved. Please discuss the necessary measures to reduce melamine exposure from environment. You can make a figure to emphasise in the most relevant way the potential sources of melamine contamination. Please discuss also the role of certain natural products or plant extract supplementation for reducing the intensity of oxidative stress and protecting the kidneys ( https://pubmed.ncbi.nlm.nih.gov/31686333/ )
Reviewer 4 Report
The study characterized the interplay between genetic polymorphism of MnSOD and the risk of renal injury in calcium urolithiasis patients underwent to melamine environmental exposure, providing evidence on the possible role of genetic factors in their increased susceptibility to oxidative damage. Subject matter is of interest and overall, the paper is well written. Hypothesis is clearly presented and it is supported by text, procedures are clear, concise, and easily replicable. Although the results are logically presented, and answer to research hypothesis, they should be revised to improve readability. The discussion need minor revision to correlate with data and link with goals. Too many tables, but can be addressed by authors.
